# Evaluation of Secondhand Smoke Using PM2.5 and Observations in a Random Stratified Sample in Hospitality Venues from 12 Cities

**DOI:** 10.3390/ijerph16081381

**Published:** 2019-04-17

**Authors:** Bekir Kaplan, Asli Carkoglu, Gul Ergor, Mutlu Hayran, Xisca Sureda, Joanna E Cohen, Ana Navas-Acien

**Affiliations:** 1Institute for Global Tobacco Control, Johns Hopkins Bloomberg School of Public Health, Baltimore, MD 21205, USA; jcohen@jhu.edu; 2Department of Psychology, Kadir Has University, Istanbul 34083, Turkey; asli.carkoglu@khas.edu.tr; 3Department of Public Health, Faculty of Medicine, Dokuz Eylul University, Izmir 35220, Turkey; gul.ergor@deu.edu.tr; 4Department of Preventive Oncology, Hacettepe University Cancer Institute, Ankara 06230, Turkey; kmhayran@gmail.com; 5Social and Cardiovascular Epidemiology Research Group, School of Medicine, University of Alcalá, Alcalá de Henares, 28801 Madrid, Spain; francisca.sureda@uah.es; 6Department of Environmental Health Sciences, Columbia University Mailman School of Public Health, NY 10032, USA; an2737@cumc.columbia.edu

**Keywords:** secondhand smoke, PM2.5, Turkey, hospitality venue

## Abstract

Background: Turkey passed a law banning smoking in all indoor public places in 2008. In response to the indoor smoking restriction, many smokers may have relocated to outdoor areas of venues. The aim of this study was to evaluate air pollution related to SHS exposure in indoor and outdoor areas of hospitality venues in 12 cities in Turkey. Method: In this cross-sectional study, we evaluated hospitality venues in 12 cities in Turkey. In each visited venue, we evaluated a pre-specified number of study locations such as the outdoor area of the main entrance, indoor areas, and patios or other outdoor dining areas, completely or partially covered with window walls. We measured particulate matter 2.5 (PM2.5) in those areas. Results: The fieldworkers visited 72 randomly selected hospitality venues and measured PM2.5 concentrations in 165 different locations (indoor, outdoor, and patios) of those venues. Overall, 2573 people were observed, 909 of them smoking. The median (IQR) PM2.5 concentrations were 95 (39–229) μg/m^3^ indoors, 25 (13–48) μg/m^3^ outdoors, and 31 μg/m^3^ (16–62) in the patios (*p* < 0.001). After adjustment, each additional smoker was associated with a 2% increase in PM2.5 concentrations in patio air (GMR (95% CI): 1.02 (1.00, 1.05), and a 4% increase in indoor air (GMR (95% CI): 1.04 (1.02, 1.05). Conclusions: There were unhealthy levels of smoking-caused PM2.5 concentrations, not only indoors, but also in the patios of hospitality venues. Legislative efforts to expand the smoke-free legislation to outdoor areas adjacent to indoor public places and an action plan to increase compliance with the smoke-free policy are urgently needed in Turkey.

## 1. Introduction

Secondhand smoke (SHS) exposure is a major cause of morbidity and mortality [1]. A third of nonsmoking adults are exposed to SHS worldwide [1]. To eliminate tobacco smoking in all indoor public places, including workplaces, Article 8 of the World Health Organization (WHO) Framework Convention on Tobacco Control (FCTC) calls for comprehensive smoke-free legislation [2,3]. In 2007, a revision of the FCTC Article 8 guidelines further recommended that parties to the FCTC prohibit smoking in outdoor or quasi-outdoor places where appropriate, based on evidence as to the possible health hazards [4].

Recently, some countries have extended smoking bans to some outdoor locations, particularly in healthcare centers and settings where children are present [5]. In many outdoor locations, however, people may be exposed to SHS, such as terraces and patios in hospitality venues and near entrances to smoke-free buildings [5]. Turkey ratified the FCTC in 2004 [6] and passed a law in 2008 banning smoking in all indoor public places that was expanded to bars, cafes, and restaurants [7]. Turkey, similar to most other countries, has not regulated smoking in outdoor hospitality venue areas.

A multicenter study carried out in hospitality venues of eight European countries reported that indoor settings where smoking is banned but which have a semi-closed outdoor area have higher levels of SHS than those with open outdoor areas, making semi-closed outdoor areas of hospitality venues a growing public health concern with respect to SHS exposure [8]. Smoking in outdoor areas of venues can reach inside and is also an occupational hazard for employees and non-smoking customers, including children, who spend time in those areas [9,10,11,12]. Evaluating outdoor public places is critical as outdoor smoking has become an emerging source of secondhand smoke exposure. Relatively few studies, however, have conducted objective measures of SHS exposure in outdoor hospitality areas [8,13,14].

In Turkey, SHS exposure has been studied in different indoor locations, especially in hospitality venues, including measures of particulate matter 2.5 (PM2.5) [15,16,17]. SHS impacts on air quality in outdoor areas of public venues, however, has not been evaluated in Turkey. While a recent 12-city study of hospitality venues randomly selected found that smoking was common in outdoor areas [15], it is unclear whether SHS emissions dilution in outdoor air represent a relevant source of pollution exposure to workers and non-smoking customers. The aim of this study was to evaluate air pollution related to SHS in indoor and outdoor areas of bars and nightclubs in 12 cities in Turkey.

## 2. Method

### 2.1. Study Population

In this cross-sectional study, we studied hospitality venues in 12 major cities (Istanbul, Ankara, Izmir, Adana, Balikesir, Bursa, Erzurum, Gaziantep, Kayseri, Samsun, Trabzon, and Van) in Turkey. Within the urban districts of each city, the Turkish Statistical Institute randomly selected 10 sampling points (a starting point on the street in each sampling area). Around each sampling point, our fieldworkers visited the closest hospitality venues such as restaurants, bars/nightclubs, cafés, and traditional coffee houses. The fieldworkers gradually expanded the search until one or two of each type of hospitality venue had been located around each sampling point and a pre-specified target number of venues of each type had been located in each study city. The target numbers, which had been set by a consensus panel before the fieldwork began, took into account the size of the city, the rarity of the type of venue, and the allocated fieldwork duration—two weeks in each major city and 1 week in the smaller cities. Overall, 884 public venues were visited for the original observational study. Slightly less than 10% (*n* = 72) were randomly selected to measure PM2.5 concentrations following a standardized protocol. The fieldworkers visited these 72 hospitality venues and measured PM2.5 concentration in a total of 165 different locations including indoor, outdoor, and the patio (if present). All venues allowed public access. The fieldwork was conducted in the winter season (December 2012 to January 2013) in Ankara, Istanbul, and Izmir and in the summer season (May to July 2013) in the other cities. Institutional review boards at the Johns Hopkins University in Baltimore (United States of America) and at Doğuş University in Istanbul (Turkey) approved the study protocol.

### 2.2. Data Collection

Following a standardized protocol, trained fieldworkers conducted all the observations and PM2.5 measures working in pairs and visiting each study venue during the venue’s rush hours in evenings between 08:00 and 00:30. In each visited venue, the fieldworkers followed a standard itinerary and evaluated a pre-specified number of study locations. In the visited venues, the locations included—when present—the outdoor area of the main entrance, indoor areas (the inside of the main entrance, indoor dining area, indoor—when present—dancing and bar areas, and bathrooms) and patios (outdoor dining areas, completely or partially covered with window wall, if present).

In each study location, the fieldworkers recorded the number of people present, the number of people smoking, the presence of “No Smoking” signs, and the presence of candles. The fieldworkers also recorded the presence of mechanic ventilation in the indoor areas and whether cigarettes were sold in the venue. For each sampling location, the exact date and time were recorded. As the legislation on the prohibition of smoking in Turkey did not apply to outdoor areas at the time of the fieldwork, any sign posted at the entrance to a venue was assumed to apply to the venue’s indoor locations.

In each of the 72 randomly selected hospitality venues, we used a SidePak AM510 personal aerosol monitor (TSI, Shoreview, MN, USA) to measure air concentrations of particulate matter with a diameter of less than 2.5 μm (PM2.5). We measured PM2.5 for 5 min outside the venue, for 20 min in the main bar area, for 5 min on the patio if present and, finally, for 5 min outside the venue again [13,18,19]. PM2.5 concentrations were corrected for relative humidity based on humidity data for each city and date, available from the city Airport. We adjusted continuous PM2.5 measurements when the relative humidity exceeded 60%, as described previously [15]. For each sampling location, the number of people and smokers and the exact date and time that the air monitoring was started and finished were recorded. The measures outside the venue at the beginning and end of the observations were combined into one single “outdoor” measure (this measure excluded the patios, which were analyzed separately unless specified).

### 2.3. Data Analysis

We determined the percentage of presence of “No Smoking” signs, candles, mechanic ventilation, and cigarettes sales overall and by size of the cities (large for İstanbul, Ankara, İzmir and small for the other cities). We also presented the number of people observed and number of smokers observed by location. We used Fisher’s exact test to compare percentages between the three larger study cities and the smaller cities.

PM2.5 concentrations were right-skewed; therefore, medians and geometric means were used for analysis. We first examined the correlation between PM2.5 and the number of people and smokers present using Spearman rank correlation coefficients. We compared the distribution of PM2.5 concentrations (median and interquartile range) by season, the presence of “No Smoking” signs and candles, and number of smokers using Mann Whitney U and Kruskal–Wallis tests.

We calculated the geometric mean ratio and 95% confidence intervals of PM2.5 concentrations per smoker present using multivariable linear regression for indoor and outdoor areas separately. The models were run with PM2.5 as log-transformed and the geometric mean ratios were obtained by exponentiating the β coefficient. Ratios were adjusted sequentially by the following a priori potential confounders: Season, presence of candle, presence of window or door opening outdoor, and number of smokers. All analyses were performed using Stata version 13.1 (StataCorp. LP, College Station, TX, USA).

## 3. Results

The fieldworkers’ observations, made in a total of 72 venues, covered 72 indoor locations—in which 1749 persons and 577 smokers were observed; 72 outdoor locations (excluding patios)—in which 171 persons and 61 smokers were observed; and 21 patio locations—in which 653 persons and 271 smokers were observed. Overall, 2573 people were observed, 909 of them smoking.

Of the 72 venues, 31 (43.1%) were in Istanbul, Ankara, and Izmir. A total of 22 (30.6%) venues had no “No Smoking” signs in at least one location of the venue (referring to the indoor smoking ban). We observed candles and mechanic ventilation in 17 (23.6%) and 35 (50.0%) venues, respectively. Cigarette sales were only observed in 1 (1.4%) venue (Table 1).

Overall, we observed people smoking in 69.7% of the venues (outdoor 61.1%, indoor 70.8%, and patio 95.2%). The percentage of venues where smoking was observed was significantly higher in the patios than in indoor and outdoor areas (*p* = 0.01).

The median (IQR) PM2.5 concentrations were 95 (39–229) μg/m^3^ indoors, 25 (13–48) μg/m^3^ outdoors (excluding the patios), and 31 (16–62) μg/m^3^ in patios. The average indoor PM2.5 concentrations were markedly higher than the outdoor and patio PM2.5 concentrations (*p* < 0.01). Although the PM2.5 concentrations were higher in patios than the other outdoor areas, the difference was not statistically significant (*p* = 0.52) (Table 2).

PM2.5 concentrations differed markedly by the number of smokers in indoor locations. In the absence of smokers, the median (IQR) indoor air PM2.5 concentrations were 37 (20–110) μg/m^3^. In the presence of 1–10, 11–20 and over 20 smokers in indoor areas, the median (IQR) PM2.5 concentrations were 82 (10–100) μg/m^3^, 195 (89–641) μg/m^3^ and 572 (377–677) μg/m^3^, respectively (*p* for trend <0.01) (Table 2).

Indoors, the median (IQR) PM2.5 concentrations were 149 (47–377) μg/m^3^ in winter, 53 (133–147) μg/m^3^ in summer. The average indoor PM2.5 concentrations were markedly higher in winter in the three largest cities than summer in the nine other cities (*p* = 0.01). PM2.5 concentrations also differed by the presence of doors and windows open outside. In the presence of doors and windows open outside, the median (IQR) PM2.5 concentrations were 53 (32–118) μg/m^3^ and 41 (23–67) μg/m^3^, respectively. In the absence of doors and windows open outside, the median (IQR) PM2.5 concentrations were 149 (46–458) μg/m^3^ and 148 (56–356) μg/m^3^, respectively (*p* < 0.01 and *p* < 0.01) (Table 2).

Indoors, PM2.5 concentrations were moderately correlated with the number of smokers (Spearman ρ = 0.43, *p* < 0.01). The correlation between the number of smokers and PM2.5 in patios was of the same magnitude as in indoor areas (Spearman ρ = 0.48, *p* = 0.03). In other outdoor areas, the correlation was weaker and non-significant (Spearman ρ= 0.17, *p* = 0.72) (Table 3).

After adjustment, each additional smoker was associated with a 4% increase in indoor air PM2.5% (GMR (95% CI): 1.04 (1.02, 1.05) and with a 2% increase in patio air (GMR (95% CI): 1.02 (1, 1.05) (Table 4).

## 4. Discussion

In this national cross-sectional study, the presence of smoking both indoors and outdoors (especially in patios/terraces) was associated with higher airborne PM2.5 concentrations. While the concentrations of PM2.5 concentrations outdoors were lower than indoors, smoking outdoors was very common and resulted in elevated PM2.5 exposure to nearby customers and employees. Each additional smoker was associated with a 2% increase in PM2.5 concentrations in patio air, and a 4% increase in indoor air. To the best of our knowledge, this is the first study measuring PM2.5 concentrations in patio and outdoor locations of hospitality venues in Turkey. The study, moreover, used a random strategy to sample hospitality venues, which is a major strength as most previous studies assessing PM2.5 concentrations in hospitality venues, in Turkey and in other countries, have relied on convenience sampling.

We recently reported the results of a systemic evaluation of public buildings in which we observed smokers, cigarette butts, and ashtrays to evaluate SHS exposure in 12 cities in Turkey [15]. Smoking was widespread in outdoor areas near entrances and in patios/gardens in most public places evaluated, including outdoor areas of hospitals, universities, malls, coffee/tea houses, and bars/nightclubs. In this study, we report the findings of measures of PM2.5, an objective measure of air quality, collected in a subset of the study samples of our former study [15]. In our former systematic observational evaluation [15], we found that compliance with the smoke-free law was low in Turkish hospitality venues. Consistent with those findings, we measure high indoor PM2.5 concentrations exceeding WHO indoor air quality standards (25 µg/m^3^ 24-h mean) [20] in the hospitality venues’ indoor and patio locations, confirming low compliance, as well as poor air quality in outdoor areas.

Smoking was observed in almost all patio locations of hospitality venues. Furthermore, the air PM2.5 concentrations in the patios where smoking was observed were higher than in outdoor areas of hospitality venues where smoking was not present. In a study measuring PM2.5 concentrations in terraces of hospitality venues in Spain [13], outdoor SHS levels were much higher in semi-closed terraces than in open outdoor areas, which was consistent with an Australian study [21]. Overall, these findings support that when individuals sit in an outdoor dining area of venues where smokers are present, they may be exposed to substantial SHS levels. The same is true for workers in those outdoor areas. The current study and previous studies [13,21], also support that banning indoor smoking, despite poor compliance in Turkey, seems to displace SHS exposure to adjacent outdoor areas such as patios and entrances.

In another study [16] measuring PM2.5 concentrations in indoor locations of hospitality venues three months before (April 2009) and four to five months after (November to December 2009) the implementation of the smoke-free ban in Turkey, the average PM2.5 concentrations significantly decreased in indoor locations of study venues after the implementation of the smoking ban. Our study, however, which was based on a random sampling strategy to select the venues and which was conducted several years after 2009, the year of the implementation of the law, found substantially high levels paired with high numbers of smokers in indoor areas. These findings indicate that exposure to SHS remains a major problem in indoor hospitality venues in Turkey. In addition to substantially improving the implementation of the legislation, there is a need to expand the smoke-free legislation to include outdoor areas of hospitality venues, in particular patios.

In February 2015, the Turkish Ministry of Health released a circular [22] proposing a ban on the use of tobacco and tobacco products in certain outdoor areas at public institutions and agencies including public outdoor areas used by children (e.g., playgrounds) or created for physical activity (e.g., walking trails, sports grounds). This circular, however, has not been implemented so far. Additional efforts are needed to improve the implementation of the current smoke-free legislation in indoor areas and to expand the smoke-free areas to outdoor places in patios and terraces of hospitality venues in Turkey.

This study has some limitations. First, we measured PM2.5 in the outdoor areas of venues. PM2.5 is not a specific marker for tobacco smoking in outdoor areas and could dissipate quickly. Our measurement could be affected by other sources of PM2.5 outdoors. Second, we measured PM2.5 for five minutes outdoors; five minutes might be too short a time to capture the real PM2.5 concentration caused by smoking, which was related to the challenge of conducting such a comprehensive observational study and fieldwork in such a large number of venues. Third, we are unable to determine if our results are representative of other small cities, towns, and communities in Turkey or whether compliance in rural areas of Turkey is similar to that which we recorded.

## 5. Conclusions

To the best of our knowledge, this is the first study measuring PM2.5 concentrations in patio and outdoor locations of hospitality venues in Turkey and the first study measuring PM2.5 indoors using a random sampling strategy to select the venues (in Turkey and in many other countries). Indoors and in patios, smoking was common and resulted in elevated PM2.5 concentrations. Presence of smoking was a major factor associated with increased PM2.5 concentrations indoors and in patios, but also outdoors. Taking into account the patio PM2.5 concentrations in this study, an outdoor smoke-free policy should accompany the indoor smoking policy in order to reduce secondhand smoke exposure in hospitality venues. Additional efforts to expand the smoke-free legislation to outdoor public areas of hospitality venues and an action plan to increase compliance with the smoke-free policy are needed to protect workers and customers effectively from SHS exposure in hospitality venues in Turkey.

## Figures and Tables

**Table 1 ijerph-16-01381-t001:** Venue characteristics by location.

Venue Characteristics	All Venues (N: 72)	Indoor (n:72)	Outdoor (n:72)	Patio (n:21)	*p* *
Season	n (%)	n (%)	n (%)	n (%)	
Winter (Istanbul, Ankara, Izmir)	31 (43.1)	31 (43.0)	31 (43.0)	9 (42.9)	1.000
Summer (9 smaller cities)	41 (56.9)	41 (57.0)	41 (57.0)	12 (57.1)	
“No Smoking” Sign					
Yes	50 (69.4)	50 (69.4)	1 (1.4)	1 (4.8)	<0.001
No	22 (30.6)	22 (30.6)	71 (98.6)	20 (95.2)	
Presence of Candle					
Yes	17 (23.6)	17 (23.6)	17 (23.6)	5 (23.8)	1.000
No	55 (76.4)	55 (76.4)	55 (76.4)	16 (76.2)	
Mechanic Ventilation During Sampling					
Yes	35 (50.0)	35 (50.0)	NA	NA	NA
No	35 (50.0)	35 (50.0)	NA	NA	
Presence of Doors Open Outside					
Yes	33 (45.8)	33 (45.8)	NA	NA	NA
No	39 (54.2)	39 (54.2)	NA	NA	
Presence of Windows Open Outside					
Yes	24 (33.3)	24 (33.3)	NA	NA	NA
No	48 (66.7)	48 (66.7)	NA	NA	
Cigarettes Sale					
Yes	1 (1.4)	NA	NA	NA	NA
No	71 (98.6)	NA	NA	NA	
People in Venues	All Locations (N:165)	Indoor (n:72)	Outdoor (n:72)	Patio (n:21)	
No. of People Observed (Med, IQR)	9 (2, 20)	19 (11, 30)	2 (1, 3)	17 (10, 40)	<0.001
No. of Smokers Observed (Med, IQR)	1 (0, 7.5)	5 (0, 11)	1 (0, 1)	9 (4, 16)	<0.001
Observed Smoking					
Yes	115 (69.7)	51 (70.8)	44 (61.1)	20 (95.2)	0.011
No	50 (30.3)	21 (29.8)	28 (38.9)	1 (4.8)	
Number of Smokers					
None	50 (30.5)	21 (29.6)	28 (38.9)	1 (4.8)	<0.001
1–10	85 (51.8)	28 (39.4)	44 (61.1)	13 (61.9)	
11–20	21 (12.8)	16 (22.5)	0 (0.0)	5 (23.8)	
>20	8 (4.9)	6 (8.5)	0 (0.0)	2 (9.5)	

* *p*-values are chi-square test of independence for proportions, and Kruskal–Wallis differences for continuous variables between locations.

**Table 2 ijerph-16-01381-t002:** Median (IQR) PM2.5 concentrations (µg/m^3^) by location among hospitality venues in Turkey.

Characteristics	N (72)	Indoor	Outdoor	Patio
All	n (%)	95 (38.5, 229) *	25 (13, 48) **	31 (16, 62)
Season				
Winter (Istanbul, Ankara, Izmir)	31 (43.1)	149 (47, 377)	20 (12, 37)	38 (31, 64)
Summer (9 smaller cities)	41 (56.9)	53 (33, 147)	28 (15, 68)	16 (9, 48)
*p*-value (a)		0.01	0.13	0.08
“No Smoking” Sign Indoors				
Yes	50 (69.4)	105.5 (45, 215)	94 (94, 94)	143 (143, 143)
No	22 (30.6)	70 (35, 373)	24 (13, 44)	30 (15, 56)
*p*-value (a)		0.93	0.17	0.10
Presence of Candle				
Yes	17 (23.6)	109 (41, 228)	20 (14, 63)	16 (16, 31)
No	55 (76.4)	83 (35, 229)	26 (13, 44)	34 (15, 63)
*p*-value (a)		0.89	0.78	0.62
Presence of Doors Open Outside				
Yes	33 (45.8)	53 (32, 118)	NA	NA
No	39 (54.2)	149 (46, 458)	NA	NA
*p*-value (a)		<0.01		
Presence of Windows Open Outside				
Yes	24 (33.3)	41 (23, 67)	NA	NA
No	48 (66.7)	148 (56, 359)	NA	NA
*p*-value (a)		<0.01		
Number of Smokers	n:164 (All Locations)			
None	50 (30.5)	37 (20, 110)	17 (12, 35)	9 (9, 9)
1–10	85 (51.8)	82 (49, 151)	29 (15, 59)	28 (13, 50)
11–20	21 (12.8)	195 (89, 641)	-	38 (16, 106)
>20	8 (4.9)	572 (377, 677)	-	49 (34, 64)
*p*-value (a)		<0.01	0.10	0.32

* *p* < 0.001 (Indoor vs. others); ** *p* = 0.52 (Patio vs. outdoor); *p*-values are Mann Whitney U or Kruskal–Wallis differences in group medians for continuous variables.

**Table 3 ijerph-16-01381-t003:** Correlations between PM2.5 concentration and number of people and number of smokers among hospitality venues by location in Turkey.

	PM2.5											
	All Areas			Indoor			Outdoor			Patio		
Variables	n	ρ *	*p*	n	ρ *	*p*	n	ρ *	*p*	n	ρ *	*p*
Number of Smokers	164	0.43	<0.01	71	0.52	<0.01	72	0.17	0.15	21	0.48	0.03

* Spearman Correlation.

**Table 4 ijerph-16-01381-t004:** Ratio of geometric means of PM2.5 concentration by venue characteristics in hospitality venues in Turkey.

Variables	n	Indoor	Outdoor	Patio
Season		GMR * (95% CI)	GMR (95% CI)	GMR * (95% CI)
Summer (Ref)	94	1	1	1
Winter	71	1.21 (0.67, 2.21)	0.68 (0.42, 1.11)	1.91 (0.81, 4.48)
No. of Smokers	164	1.04 (1.02, 1.07)	1.09 (0.86, 1.38)	1.02 (1.00, 1.05)

* GMR adjusted by presence of burning source, presence of window, or door opening outdoors.

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
