# Peer review of "Evaluation of Secondhand Smoke Using PM2.5 and Observations in a Random Stratified Sample in Hospitality Venues from 12 Cities"

_ijerph, 2019, doi:10.3390/ijerph16081381_

Round 1

Reviewer 1 Report

The study presented in this paper aims to evaluate the indoor and outdoor air quality, in terms of PM2.5, of 72 hospitality venues of 12 major cities in Turkey. Secondhand smoke exposure is a major health issue, therefore the relevance of the paper is justified. I would like to highlight the accurate random sampling strategy that supports the validity of the results found here. Nevertheless, my main concern is that this study lacks of blank samples to link PM2.5 concentrations with smoking without bias. Blank samples could be smoke-free hospitality venues, patios and outdoor spaces. As the author describe in the study limitations section (lines 219-226), PM2.5 is not a specific marker of tobacco smoke, and other sources such as road traffic or biomass burning (relevant in winter) influence its concentration. Therefore, blank samples are highly desirable for this monitoring studies. I recommend the authors, at least check the official PM2.5 measurements outdoors of each city and include these values in the discussion.

Minor specific comments:

1.    Title: I have a subtle objection about the first part of the title: “Smoke-free Turkey”, that suggests that in some ways, Turkey is smoke-free. However, the results of this study show that Turkey is not smoke-free, not at all, since none of the selected hospitality venues complies the law from 2008 that ban smoking in indoor public places. I suggest the authors to eliminate that part of the title or at list add a question mark.

2.    Abstract: In the conclusions section, please remark that law enforcement is indoor places is needed.

3.    Lines 123-126: this paragraph is confusing, please revise the punctuation marks. In line 123 “covered 341 indoor locations”. Is 341 right? According with Table 1, 72 indoor locations were studied (31+41).

4.    I suggest to restructure Table 1. The first two sections provide information about the venues (n (%)), followed by two more sections about the people in the venues (med, IQR) and next 4 more sections about the venue characteristics (n (%)). This mixed information is hard to follow by the reader. Split the table in two (one about venue characteristics and another one about people in venues) may help. Also, please reduce redundant information from this Table. For instance, “No smoking” sign: “Yes” 50 (69.4) (if the total venues are 72, “No” are obviously 22 venues).

5.    Line 151: Please add the concentration units (µg/m3) to Table 2 caption.

6.    Table 2: some associations of PM2.5 concentrations and venue characteristics are not discussed in the text, please add a short paragraph.

7.    Table 2 & 3: Please clarify the Number of smokers (n=164). In line 126, the authors stated that the number of smokers was a total of 909. Please, indicate why n=164 has been used for the statistical calculations.

8.    Conclusions, same as comment 2: remark the breach of law in indoor spaces.

Author Response

Reviewer 1

The study presented in this paper aims to evaluate the indoor and outdoor air quality, in terms of PM2.5, of 72 hospitality venues of 12 major cities in Turkey. Secondhand smoke exposure is a major health issue, therefore the relevance of the paper is justified. I would like to highlight the accurate random sampling strategy that supports the validity of the results found here. Nevertheless, my main concern is that this study lacks of blank samples to link PM2.5 concentrations with smoking without bias. Blank samples could be smoke-free hospitality venues, patios and outdoor spaces. As the author describe in the study limitations section (lines 219-226), PM2.5 is not a specific marker of tobacco smoke, and other sources such as road traffic or biomass burning (relevant in winter) influence its concentration. Therefore, blank samples are highly desirable for this monitoring studies. I recommend the authors, at least check the official PM2.5 measurements outdoors of each city and include these values in the discussion.

Response: We appreciated this important feedback that has helped us to improve the content of our article. In Turkey, the National Air Quality Monitoring Network measures the air quality at the national level. However, they use PM10 concentrations in order to measure air quality. In addition, we could not find comparable PM2.5 results including 2013 calendar year at the city level. Therefore, we are unable to present a comparable blank sample for this study. We have added the following to the discussion to acknowledge that this type of data would have been helpful. The changes are as follows (Page 7 Line 224): “Using a blank sample to compare our PM2.5 results by city would be valuable to take into account background levels of PM2.5. However, the official air quality measures use PM10. Therefore, we were unable to compare our PM2.5 results with background levels.”

Minor specific comments:

1. Title: I have a subtle objection about the first part of the title: “Smoke-free Turkey”, that suggests that in some ways, Turkey is smoke-free. However, the results of this study show that Turkey is not smoke-free, not at all, since none of the selected hospitality venues complies the law from 2008 that ban smoking in indoor public places. I suggest the authors to eliminate that part of the title or at list add a question mark.

Response: “Smoke-free Turkey” has been dropped from the title.  The new title of the manuscript is “Evaluation of Secondhand Smoke Using PM2.5 and Observations in a Random Stratified Sample in Hospitality Venues from 12 Cities.”

2. Abstract: In the conclusions section, please remark that law enforcement is indoor places is needed.

Response: A statement about the enforcement of the law was added to the abstract conclusion. The changes are as follows (Page 1 Line 30): “Legislative efforts to expand the smoke-free legislation to outdoor areas adjacent to indoor public places and an action plan to increase compliance with the smoke-free policy are urgently needed in Turkey.”

3. Lines 123-126: this paragraph is confusing, please revise the punctuation marks. In line 123 “covered 341 indoor locations”. Is 341 right? According with Table 1, 72 indoor locations were studied (31+41).

Response: The number 341 refers to the total number of different areas of indoor locations such as bar area, bathroom, and main dining areas. However, we realize that this may cause confusion. Therefore, we have replaced 341 with 72 which represents the total number of indoor locations.

4. I suggest to restructure Table 1. The first two sections provide information about the venues (n (%)), followed by two more sections about the people in the venues (med, IQR) and next 4 more sections about the venue characteristics (n (%)). This mixed information is hard to follow by the reader. Split the table in two (one about venue characteristics and another one about people in venues) may help. Also, please reduce redundant information from this Table. For instance, “No smoking” sign: “Yes” 50 (69.4) (if the total venues are 72, “No” are obviously 22 venues).

Response: The table was updated  in the attachment 

5. Line 151: Please add the concentration units (µg/m3) to Table 2 caption.

Response: Concentration unit (µg/m3) was added to the Table 2 caption.

6.       Table 2: some associations of PM2.5 concentrations and venue characteristics are not discussed in the text, please add a short paragraph.

Response: The paragraph below was added to summarize the data from table 2 (Page 5 Line 149):

Indoors, the median (IQR) PM2.5 concentrations were 149 (47–377) μg/m3 in winter, 53 (133-147) μg/m3 in summer. The average indoor PM2.5 concentrations were markedly higher in winter in the three largest cities than summer in the nine other cities (p=0.01). PM2.5 concentrations also differed by the presence of doors and windows open outside. In the presence of doors and windows open outside, the median (IQR) PM2.5 concentrations were 53 (32-118) μg/m3 and 41 (23-67) μg/m3, respectively. In the absence of doors and windows open outside, the median (IQR) PM2.5 concentrations were 149 (46-458) μg/m3 and 148 (56-356) μg/m3, respectively (p<0.01 and p<0.01) (Table 2).”

7.       Table 2 & 3: Please clarify the Number of smokers (n=164). In line 126, the authors stated that the number of smokers was a total of 909. Please, indicate why n=164 has been used for the statistical calculations.

Response: The number 164 refers to the total number of locations in all venues (indoor+outdoor+patio). In order to eliminate confusion, we added the statement “All Locations” next to this value in the Table 2 and “n (%)” on the top of this column.

8.       Conclusions, same as comment 2: remark the breach of law in indoor spaces.

Response: A remark was added to the conclusion as follows (Page 7 Line 233):

Additional efforts to expand the smoke-free legislation to outdoor public areas of hospitality venues and an action plan to increase compliance with the smoke-free policy are needed to protect workers and customers effectively from SHS exposure in hospitality venues in Turkey.”

Reviewer 2 Report

The introduction is a good review on the current knowledge.

The methods are described in details.

Authors presented their results systematically and illustrated them with tables that are helpful in the text reading.

The paper is written in coherent style with use of a proper terminology.

I highly evaluate a talent for clear summary and interpretation of results on the background of the current literature.

In summary I can state that the aim of the study was achieved.

Author Response

Reviewer 2 The introduction is a good review on the current knowledge. The methods are described in details. Authors presented their results systematically and illustrated them with tables that are helpful in the text reading. The paper is written in coherent style with use of a proper terminology. I highly evaluate a talent for clear summary and interpretation of results on the background of the current literature. In summary I can state that the aim of the study was achieved. Response: We thank the reviewer for the positive comments about our manuscript.

Reviewer 3 Report

This study looks at air pollution related to outdoor SHS at hospitality venues in Turkey. This is an important area of research given that many localities are implementing indoor smoking bans and because outdoors is a common source of SHS exposure among non-smokers. Overall, the paper is well written, clear, and provides an important contribution to the literature. However, there are a handful of ways in which the paper could be strengthened. 

Method

Can you define what a "sampling point" is? (line 66)

Provide justification for why outdoor areas including patios were combined for this one analysis but kept separate for others (line 117). 

Results

What is the logic behind the association between number of people (smokers and non-smokers) and PM2.5 concentration? (line 155) Is there existing literature that suggests that this might be the case? Need more justification for why that's being examined.

Just a comment...It is odd that the association between smokers and PM2.5 in patios and other outdoor areas was so different. results for patios looked more like results for indoors as opposed to outdoors. This touches on my previous point about patios being combined with outdoors (as opposed to being kept separate). This analysis suggests that, from a statistical standpoint, patios may look more like indoor areas.

So, in Table 3, every single person observed in each area was a smoker? Am I understanding that correctly? Or, were there an identical number of people (non-smokers + smokers) in each location? Or is this a typo? 

In Table 4, I'm assuming the n (94+71) should sum to 164 (not 165).

Discussion

I think it is important to put the PM2.5 values in context. You mention that they were high in line 185, but what is high? What's "safe"? You could address this by adding a sentence or two in the intro and/or citing literature that quantifies the harmful effects of exposure to different levels of PM2.5. 

You state that PM2.5 was higher in patios in winter than in summer (line 198). I didn't see this finding presented in the text in the results section, and in Table 4, it looks like the 95% CI for winter firmly includes 1.00 (.81, 4.48). Is the difference by season for patios significant? 

One thing to consider noting is that, if compliance/enforcement of the indoor smoking ban does improve, then it could increase the need for an outdoor no-smoking policy. Perhaps you recommend that these two initiatives move forward in tandem.

You mention limitations of the methodology (line 222). Do you have references/justification for why that amount of time was chosen in the first place? Have other studies used those methods with success?

Line 225-226, I understand that you wouldn't be able to generalize to rural areas, but your methods seem to suggest that you could generalize to other major cities in Turkey, no?

Author Response

Reviewer 3

This study looks at air pollution related to outdoor SHS at hospitality venues in Turkey. This is an important area of research given that many localities are implementing indoor smoking bans and because outdoors is a common source of SHS exposure among non-smokers. Overall, the paper is well written, clear, and provides an important contribution to the literature. However, there are a handful of ways in which the paper could be strengthened.

Method

Can you define what a "sampling point" is? (line 66)

Response: Sampling points refer to a starting point on the street in each sampling area which were selected randomly by the Turkish Statistical Institute. The changes are as follows (Page 2 Line 66): “the Turkish Statistical Institute randomly selected 10 sampling points (a starting point on the street in each sampling area)”

Provide justification for why outdoor areas including patios were combined for this one analysis but kept separate for others (line 117).

Response: Thank you for this important feedback. We have dropped the analyses that present the all outdoor (outdoor+patio) results from Tables 2, 3, and 4, and from the text as well.

What is the logic behind the association between number of people (smokers and non-smokers) and PM2.5 concentration? (line 155) Is there existing literature that suggests that this might be the case? Need more justification for why that's being examined.

Response: Thank you for this feedback. We have dropped the association between number of people and PM2.5 concentrations from the text and Table 3 as well.

Just a comment...It is odd that the association between smokers and PM2.5 in patios and other outdoor areas was so different. results for patios looked more like results for indoors as opposed to outdoors. This touches on my previous point about patios being combined with outdoors (as opposed to being kept separate). This analysis suggests that, from a statistical standpoint, patios may look more like indoor areas.

Response: Thank you for this important feedback. As mentioned earlier, we have dropped the column that presents the all outdoor (outdoor+patio) results from Tables 2, 3, and 4, and from the text as well.

So, in Table 3, every single person observed in each area was a smoker? Am I understanding that correctly? Or, were there an identical number of people (non-smokers + smokers) in each location? Or is this a typo?

Response: In the first row (number of people) we count all people (smokers+nonsmokers) observed. In the second row, we only counted number of smokers observed. We also dropped the association between number of people and PM 2.5 concentration from the table and text as well.

In Table 4, I'm assuming the n (94+71) should sum to 164 (not 165).

Response: In the regression model, 164 people were included for the variable “No. of Smokers.” In the season variable, there are 94 people in the summer group and 71 people in the winter group.

Discussion

I think it is important to put the PM2.5 values in context. You mention that they were high in line 185, but what is high? What's "safe"? You could address this by adding a sentence or two in the intro and/or citing literature that quantifies the harmful effects of exposure to different levels of PM2.5.

Response: We have added the WHO air quality standard to the relevant sentence. The changes are as follows (Page 6 Line 185): “In our former systematic observational evaluation [15] we found that compliance with the smoke-free law was low in Turkish hospitality venues. Consistent with those findings, we measured high indoor PM2.5 concentrations exceeding WHO indoor air quality standards (25 µg/m3 24-hour mean) [18] in the hospitality venues’ indoor and patio locations, confirming low compliance with the legislation, as well as poor air quality in outdoor areas.” [18] WHO. Air Quality Guidelines: Global update 2005. Copenhagen: WHO Regional Office for Europe, 2006. https://apps.who.int/iris/bitstream/handle/10665/69477/WHO_SDE_PHE_OEH_06.02_eng.pdf;jsessionid=928B76B3801EA45AA7A022312DC74217?sequence=1 You state that PM2.5 was higher in patios in winter than in summer (line 198).

I didn't see this finding presented in the text in the results section, and in Table 4, it looks like the 95% CI for winter firmly includes 1.00 (.81, 4.48). Is the difference by season for patios significant?

Response: Thank you for this feedback. We had considered the univariate analysis (Table 2) for our statement. However, the relationship did not remain significant in the regression. Therefore, we have dropped that paragraph from the discussion.

One thing to consider noting is that, if compliance/enforcement of the indoor smoking ban does improve, then it could increase the need for an outdoor no-smoking policy. Perhaps you recommend that these two initiatives move forward in tandem.

Response: We have added one sentence to the conclusion in response to this helpful advice. The changes are as follows (Page 7 Line 235): “Taking into account the patio PM2.5 concentrations in this study, an outdoor smoke-free policy should accompany the indoor smoking policy in order to reduce secondhand smoke exposure in hospitality venues.”

You mention limitations of the methodology (line 222). Do you have references/justification for why that amount of time was chosen in the first place? Have other studies used those methods with success?

Response: We added two more references to the relevant sentence, including a review of numerous studies that have conducted environmental measures of PM2.5 using similar sampling times as our study. Hyland A, Travers MJ, Dresler C, Higbee C, Cummings KM. A 32-country comparison of tobacco smoke derived particle levels in indoor public places. Tob Control. 2008 Jun;17(3):159–65. doi: http://dx.doi.org/10.1136/tc.2007.020479 PMID: 18303089 Apelberg BJ, Hepp LM, Avila-Tang E, Gundel L, Hammond SK, Hovell MF, et al. Environmental monitoring of secondhand smoke exposure. Tob Control. 2013 May;22(3):147–55. doi: http://dx.doi.org/10.1136/tobaccocontrol-2011-050301 PMID: 22949497

Line 225-226, I understand that you wouldn't be able to generalize to rural areas, but your methods seem to suggest that you could generalize to other major cities in Turkey, no?

Response: Thank you for helping us to clarify. Our results are from 12 major cities. We have added “small” to the relevant sentence.